Enhancing monitoring of suspicious activities with AI-based and big data fusion

Vorapatratorn Surapol surapol.vor@mfu.ac.th
Center of Excellence in Artificial Intelligence and Emerging Technologies, School of Information Technology, Mae Fah Luang University , Muang , Chiang Rai , Thailand
Aleem Muhammad
Electronic publication date: 2024 Jan 25
Publication date: 2024
Volume: 10
Electronic Location ID: e1741
Received 2023 Jul 26; Accepted 2023 Nov 15
Copyright: ©2024 Vorapatratorn
Copyright year: 2024
Copyright holder: Vorapatratorn
License: This is an open access article distributed under the terms of the Creative Commons Attribution License, which permits unrestricted use, distribution, reproduction and adaptation in any medium and for any purpose provided that it is properly attributed. For attribution, the original author(s), title, publication source (PeerJ Computer Science) and either DOI or URL of the article must be cited.
License URL: https://creativecommons.org/licenses/by/4.0/

Keywords: Big data, Machine learning, Classification, Web application, Data warehouse, Hadoop hive

Funding: The Office of the Ministry of Higher Education, Science, Research, and Innovation The research received funding from the Office of the Ministry of Higher Education, Science, Research, and Innovation. The funders had no role in study design, data collection and analysis, decision to publish, or preparation of the manuscript.

==============================
This study provides an AI-based detection tool for the surveillance of suspicious activities using data fusion. The system leverages time, location, and specific data pertaining to individuals, objects, and vehicles associated with the agency. The study’s training data was obtained from Thailand’s military institution. The study focuses on comparing the efficiency between MySQL and Apache Hive for big data processing. According to the findings, MySQL is better suited for quick data retrieval and low storage capacity, while Hive demonstrates higher scalabilities for larger datasets. Furthermore, the study explores the practical utilization of web applications interfaces, enabling real-time display, analysis, and identification suspicious activity results. The web application, built with NuxtJS and MySQL, includes statistics charts and maps that show the status of suspicious items, cars, and people, as well as data filtering options. The system utilizes machine-learning algorithms to train the suspicious identification model, with the best-performing algorithms being the decision tree, reaching 98.867% classification accuracy.

Introduction

The issue of armed conflict in Thailand’s southern border regions has serious consequences for national security, the protection of government personnel, and the safety of inhabitants. Moreover, it adversely affects the economy, investments and industry of tourism. The current approach does not make use of trusted technologies and information sources. Furthermore, individuals’ roles have shifted from being passive consumers of official data to active creators and disseminators of information. However, no genuine remedy to these problems has been created or implemented (Schuldt, 2021). These issues have prompted study and development efforts in the realm of surveillance technologies (Thaiparnit, Chumuang & Ketcham, 2018; Inyaem et al., 2010; Inyaem, Meesad & Haruechaiyasak, 2009). One such initiative involves a security program concept that verifies identities of individuals by analyzing their facial features before granting access to boarding areas, entrances and passenger cabins (Pumpong et al., 2021; Bite, 2010; Milne et al., 2019). Additionally, a prototype area security programming system has been developed to identify suspicious items near the base positions (Jaturapitpornchai et al., 2019; Steno et al., 2021; Kemp, 2007).

Considering these circumstances, our study proposes a detection system for suspicious activities based on artificial intelligence for organization surveillance. The system utilizes fusion of large amounts of data and aims to fulfill the requirement for comprehensive and efficient system development, with a specific focus on integrating news data from diverse databases. Analyzing and displaying connections between occurrences via charts and maps, this system is poised to effectively support big data analysis in the near future.

Literature Review

In recent years, the rising conflict in Thailand’s southern frontier areas has become an important concern, harming national security, the protection of governmental staff and the well-being of inhabitants. This issue has also led to significant economic losses, discouraging investments and impacting the tourism sector. The absence of effective technology and trusted information sources compounds the challenge faced in resolving this crisis (Ansell, Boin & Keller, 2010). Individuals have transformed from passive consumers of government data to active creators and disseminators of news and information because of the emergence of social media and citizen journalism. However, despite the abundance of information available, there is a lack of concrete applications and systems developed to address this specific problem (Baine, 2001). To bridge this gap, researchers have explored the application of surveillance technology as a potential solution.

Several studies have focused on the development and implementation of surveillance systems to enhance security measures. For instance, a prototype security program that employs facial recognition technology to verify individuals’ identities before granting access to restricted areas, such as boarding zones, entrance points and passenger cabin areas. This approach enables real-time monitoring and enhances overall security measures (Ashbourn, 2014). Apart from facial recognition, security programming systems for the area have been created to identify objects near ground and air base sites. These systems utilize advanced technologies, such as sensors and video analytics, to identify potential threats and ensure the safety of the surrounding areas (Chong & Kumar, 2003). These initiatives highlight the growing interest and investment in surveillance technology to address security concerns.

To further advance the capabilities of security monitoring systems, the integration of big data fusion has emerged as a promising approach. By leveraging diverse data sources and integrating information from various databases, the effectiveness and efficiency of security monitoring can be significantly enhanced. This integration allows for the analysis of relationships between events, which can be presented visually through graphs and maps (Cai et al., 2016). Considering the importance of data integration and analysis, artificial intelligence (AI)-based systems have garnered attention for their potential to improve security monitoring. Such systems employ machine learning algorithms to analyze large volumes of data and identify suspicious activities. In the context of agency security monitoring, an AI-powered suspicious detection system using big data fusion holds great potential in providing a comprehensive and efficient solution (Guo & Polak, 2021).

Methodology

Information for this research was obtained from the Navaminda Kasatriyadhiraj Royal Thai Air Force Academy, serving as a military college for students of the Royal Thai Air Force. The academy’s database organizes and retains data concerning the exit and entering of people, vehicles and items at various timestamps, facilitating the identification of suspicious individuals, cars and objects. The classification results derived from this dataset are displayed on a web application. Figure 1 illustrates the system diagram depicting the proposed methodologies.

Figure 1 Our proposed approaches are depicted in a system diagram.

Designing the big data infrastructure

The proposed system topology in this scholarly article, as depicted in Fig. 2, is driven by the VMware ESX server (Desai et al., 2013), serving as a virtual machine server. It includes a Web server, an Airflow server and a database server, all functioning on an Ubuntu server (LaCroix, 2018). These elements collaborate utilizing Apache Airflow (Harenslak & De Ruiter, 2021) as a data pipeline for the extract-transform-load (ETL) process, which involves extracting data from the API and transferring it to the database server for processing by the web server. While the system stores data as Big Data in the database for presentation on the web application, the careful selection of a suitable database server holds significant importance.

Figure 2 Overview our system topology.

This research utilizes Apache Hive (Bansal, Chawla & Kurle, 2019) as the server of databases. Figure 3 illustrates the essential components of the Apache Hive and Apache Hadoop architecture, which include the Metastore responsible for storing partition metadata to facilitate data verification across the entire group. Metadata and relational databases are used to store the data to ensure replication tracking and provide backups in case of data loss. The driver component oversees and monitors the progress of operations, accepting commands and storing generated metadata during the execution of HiveQL commands. After the MapReduce process is finished, the driver collects the data and initiates queries to obtain the results. The compiler then converts the command and the executor sends the HiveQL command to the MapReduce executor, enabling data processing. The Optimizer Executor communicates with Apache Hive to schedule instructions for execution in the future. Lastly, the UI, CLI and Thrift servers provide users with the capability to manage Hive through various means, such as the user interface (UI), command line (CLI), or Thrift.

Figure 3 Our Hadoop hive topology.

Training the model for suspicious activities

Importing data from a data warehouse, which contains access databases that record the positions of humans, vehicles and items, are required for training the model. A feature-extracting approach is used to transform these datasets into feature vectors and experts assign labels such as “suspicious” and “normal” (Table 1).

Table 1 The recorded data to be used in the ai training feature for suspicious prediction.

Attribute	Description	Example	
ObjectKey (ID)	Detected object reference number	001	
Time_diff	Previously detected time until
latest (in hours)	5 h	
Date_diff	Previously detected date until
latest (in days)	3 days	
Month_current	Latest detected month name	January	
Day_current	Latest detected day name	Monday	
SubDistric_current	Latest detected sub-district name	Phayathai	
District_current	Latest detected district name	Pathumwan	
Province_current	Latest detected province name	Bangkok	
SubDistric_diff	Previously detected sub-district and latest (0-same, 1-difference)	1	
District_diff	Previously detected district and latest (0-same, 1-difference)	1	
Province_diff	Previously detected province and latest (0-same, 1-difference)	1	
Location_diff	Previously detected GPS-location and latest distance (in kilometers)	417 km.	
Object_Label
(Label)	Suspicious status (0-normal,
1-suspicious)	1	

The data is separated into three sets: training, validation and testing. Figure 4 depicts the training processes used in this investigation. Data pipelines are used in the system to train automated models that extract data from many sources, analyze it and store it in a database. Apache Airflow is used to build a Python-based (Shein, 2015) data pipeline with job sequencing and scheduling. In this study, five pipelines are employed to train the automated model every day at midnight, as shown in Fig. 5.

Figure 4 Suspicious model training diagram.

Figure 5 The training data pipeline with Apache Airflow.

Classification of suspicious activities

Figure 6 depicts a schematic showing the classification of suspicious people, vehicles and objects. The feature extraction procedure, like the feature extraction approach used during training begins with the entrance and exit data of individuals, cars and items.

Figure 6 Suspicious classification diagram.

The newly collected data is then loaded into a machine-learning classifier that employs the pre-trained suspicion model. This study employs five processes to obtain categorization results at midnight each day (Fig. 7). The grouping results are recorded in a stored procedure table and then shown on the web interface.

Figure 7 The classification pipeline with Apache Airflow.

Web application UI design

The online suspect-detecting method for monitoring agency security is developed based on the requirements gathered from the Thailand Ministry of Defense, allowing authorized users to access and review suspicious data through the web application. This includes the role of reporters responsible for recording the transit of cars, items and persons inside the department. Laptops, tablets and handheld devices could all access the system.

The system’s use case diagram, depicted in Fig. 8, showcases its division into two main groups: regular users and administrators. Before to accessing the system, users are required to log in. General users are granted access to statistics concerning objects, individuals and vehicles, presented in the form of charts, statistical tables and maps. Moreover, data can be downloaded in Excel format. The system provides filtering options based on location, time and object, vehicle, or person types. On the other hand, administrators possess the same viewing capabilities as regular users, but they also enjoy additional privileges, including the ability to add, edit, or delete regular user accounts.

Figure 8 System use case diagram.

Result and Discussion

The Results and Discussion section presents the outcomes of the experiments and evaluations conducted to assess the performance of the proposed suspicious identification system. It covers the evaluation of different technologies, such as MySQL and Apache Hive, for processing big data and compares their speed and capacity. Additionally, the effectiveness of various machine learning algorithms, including ANN, SVM, k-NN, decision tree and naive Bayes, is examined for training the suspicious identification model. Furthermore, this section discusses the utilization of the web application interface, highlighting its features and functionalities for accessing and analyzing the collected data.

Performance evaluation of suspicious identification

We conducted experiments in this study to evaluate the performance of two dissimilar systems, MySQL and Apache Hive, in managing massive data for suspicious detection purposes. To conduct the experiment, we created a virtual machine using ESXI vSphere that ran the Ubuntu Server 20.04 OS and had 4 GB of memory. Hadoop 3.3.1  (Holmes, 2014) and MySQL 5.7 (Widenius, Axmark & Arno, 2002) were used as software. The data for the experiment varied from 10,000 to 5,000,000 records and was received in CSV format from the online Excel BI dataset (Excel BI Analytics, 2023). The experiment’s major purpose was to assess the processing speed and power of different technologies when dealing with massive amounts of data.

The experimental results revealed that the performance varied between MySQL and Apache Hive. MySQL demonstrated better suitability for programs requiring immediate data retrieval and smaller storage capacities. On the other hand, Hive showed better efficiency in handling larger datasets. Specifically, in terms of speed, MySQL outperformed Hive in simple queries. However, as the dataset size increased beyond 100,000 records, MySQL’s performance became unresponsive. This limitation indicates that MySQL may not be the optimal choice for processing extremely large datasets. Overall, Hive exhibited better scalability and performed well in handling big data as shown in Table 2.

Table 2 Processing time between Apache Hive and MySQL.

No. of
sample	Dataset size	Processing time (Sec.)	
		Hadoop Hive	MySQL	
10 K	358 KB	15.36	5.47	
50 K	1.77 MB	33.14	9.63	
100 K	3.54 MB	47.32	18.66	
500 K	17.68 MB	56.81	Not responding	
1 M	35.34 MB	64.37	
5 M	196.98 MB	72.14	

Furthermore, we conducted experiments to identify the most suitable machine learning algorithm for building the suspicious identification model. The model’s training data comprised of extraction of features data recorded in CSV format, comprising object, person and vehicle datasets as shown in Table 3. Figure 9 to Figure 11 depict the process of dataset acquisition.

Table 3 The number of sample and label in each dataset.

Dataset	No. of
sample	Number of label	
		0-Normal	1-Suspicious	
Object	9,950	2,450	7,500	
Person	390	230	160	
Vehicle	2,280	2,040	240	
Total	12,620	4,720	7,900	

Figure 9 The object data from CCTV.

Figure 10 The person data from face scan.

Figure 11 The vehicle data from license plate readers.

We utilized popular machine learning algorithms, including artificial neural networks (ANN) (Yegnanarayana, 2009), support vector machines (SVM) (Steinwart & Christmann, 2008), K-nearest neighbor (k-NN) (Larose & Larose, 2014), decision tree (Myles et al., 2004) and naive Bayes (Webb, Keogh & Miikkulainen, 2010), implemented using Scikit-learn in Python (Pedregosa et al., 2011). The performance of each algorithm was evaluated based on classification accuracy and prediction speed.

Table 4 summarizes the results of the algorithm’s performance, calculated as the mean outcome of 10 training rounds. Among the algorithms tested, the decision tree has the best classification accuracy (98.867%) and the fastest prediction speed (0.005 ms per sample). These findings indicate that the decision tree algorithm is well-suited for the suspicious activity identification task, offering both high accuracy and efficient prediction capabilities.

Table 4 The model performance for each AI algorithm comparison.

Algorithm	Training
time (sec.)	Predict
speed (ms.)	F1-score
(%)	Accuracy
(%)	
ANN	490.061	0.354	96.147	96.482	
SVM	35.883	0.031	82.971	82.914	
k-NN	9.988	0.135	89.947	67.315	
Decision tree	2.992	0.005	99.335	98.867	
Naïve Bayes	3.985	0.006	84.917	84.924	

Web application utilization

To provide a user-friendly interface for accessing and analyzing the suspicious identification data, we developed a web application using NuxtJS (Nuxt2, 2023), MySQL and the Python pipeline on a Windows Server 2008 (Microsoft, 2023). Users can access the web application through http://datascience.mfu.ac.th/datafusion. The application offers various features to visualize and explore the collected data.

The web application’s homepage displays historical graphs for the specified time, which is set by default to the previous 7 days. These graphs provide insights into suspect statuses, object types, vehicle types and person types, enabling users to easily grasp the overall trends and patterns. Figures 12 and 13 illustrate the visual representation of these statistical charts. Additionally, the map page, as shown in Fig. 14, displays the geographical distribution and status of suspected objects, vehicles and individuals. Different colors are used to represent varying levels of suspicion, enabling users to quickly identify areas of concern. The map presents data from the previous 7 days, allowing for temporal analysis of suspicious activities.

Figure 12 The total chart home page, first part.

Figure 13 The total chart home page, second part.

Figure 14 The suspicion classification result as show as four difference pin color in map page.

Users can utilize the search field to filter data based on location, period and suspicious status, facilitating targeted exploration of the dataset. Furthermore, the web application enables users to download the data in Excel format, providing them with the flexibility to conduct further analysis or integrate it into their own systems as shown in Fig. 15. Overall, the web application provides an intuitive and interactive interface for users to explore and understand the suspicious identification data collected. It enhances accessibility, enabling users to gain valuable insights and make informed decisions based on the presented information.

Figure 15 The raw data and download button on map page.

Summary and implications

The experimental results highlight the importance of choosing the appropriate technology for processing big data in the context of suspicious identification. MySQL proves to be efficient for applications needing immediate data retrieval and limited storage capacity, whereas Hive has superior scalability for handling bigger volumes. Moreover, the decision tree algorithm emerges as the most suitable machine learning algorithm, offering high classification accuracy and fast prediction speed for the suspicious identification task.

The developed web application complements the data processing and analysis, providing users with a user-friendly interface to access, visualize and explore the collected suspicious identification data. It facilitates effective decision-making and enhances the overall security monitoring capabilities of the agency. These findings and the implemented system have implications for enhancing agency security monitoring, enabling timely detection of suspicious activities and improving the efficiency of threat prevention efforts. The integration of big data processing, machine learning algorithms and web application interfaces contribute to the advancement of security systems and has the potential to be applied in real-world scenarios, ensuring public safety and national security.

Conclusions

This research aimed to develop an AI-powered suspicious detection solution for agency surveillance that utilizes the use of big data integration. The study successfully implemented a system that incorporates time, location and specific data related to individuals, objects and vehicles associated with the agency. The machine learning algorithms used, including artificial neural networks (ANN), K-nearest neighbor (k-NN), support vector machines (SVM), naive Bayes and decision tree were evaluated for training the suspicious identification model. The decision tree algorithm emerged as the most effective, achieving a high classification accuracy and the fastest prediction. Although the developed system showed promising results, there are limitations to consider. The study’s data were limited to the working environment of a particular organization. Generalizing the findings to other agencies requires expanding the system’s applicability and considering the unique needs of different organizations. Additionally, most of the data used in the study were simulated due to confidentiality and personal safety concerns. Future improvements should focus on incorporating real-world data to enhance the system’s efficiency and accuracy. Furthermore, the size of the dataset used in the study was limited, which may have influenced the assessment of the system architecture’s performance.

Future research should address this limitation by conducting extensive testing on larger datasets to validate the system’s scalability and effectiveness in handling big data challenges. The developed AI-based suspicious identification system offers practical implications for agency security monitoring. It provides a robust framework for detecting and identifying suspicious activities by leveraging big data fusion and machine learning algorithms. The real-time display of results on a web application enhances the efficiency of security monitoring processes, allowing for timely decision-making and threat prevention. Moreover, the system can be customized to meet the specific requirements of different agencies. The body of knowledge and guidelines derived from this research can serve as a foundation for developing similar systems tailored to the needs of diverse organizations. In conclusion, this study presents an AI-powered suspicious detection system that utilizes big data fusion and machine learning algorithms for agency security monitoring. The research successfully demonstrated the effectiveness of the decision tree algorithm in achieving high classification accuracy and fast prediction speed. While there are limitations to be addressed, such as data generalizability and the use of simulated data, the system shown great potential for enhancing security measures.

Supplemental Information

Supplemental Information 1 Dataset in MySQL Database

The training data used in this research was derived from Navaminda Kasatriyadhiraj Royal Thai Air Force Academy, a military college responsible for training Royal Thai Air Force officer cadets. Additionally, the web server used in this study was supported by the Center of Excellence in Artificial Intelligence and Emerging Technologies, School of Information Technology, Mae Fah Luang University.

Additional Information and Declarations

Competing Interests

Author Contributions

Data Availability

The authors declare there are no competing interests.

Surapol Vorapatratorn conceived and designed the experiments, performed the experiments, analyzed the data, performed the computation work, prepared figures and/or tables, authored or reviewed drafts of the article, and approved the final draft.

The following information was supplied regarding data availability:

The data is available at Zenodo: Vorapatratorn, S. (2023). Enhancing Suspicious Activity Monitoring System with AI-Based and Big Data Fusion. Zenodo. https://doi.org/10.5281/zenodo.10010468.

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
