# Peer review of "Enhancing monitoring of suspicious activities with AI-based and big data fusion"

_PeerJ Computer Science, doi:10.7717/peerj-cs.1741_

## Round 0.1 · original submission · Major Revisions

Based on the reviewers’ comments, you may resubmit the revised manuscript for further consideration. Please consider the reviewers’ comments carefully and submit a list of responses to the comments along with the revised manuscript. Once all the reviewers are satisfied with the revised version only then it can be recommended for publication.

**Language Note:** The review process has identified that the English language must be improved. PeerJ can provide language editing services - please contact us at copyediting@peerj.com for pricing (be sure to provide your manuscript number and title). Alternatively, you should make your own arrangements to improve the language quality and provide details in your response letter. – PeerJ Staff

Reviewer 1 ·

Basic reporting

The paper is somewhat well written. Although the language is appropriate and the grammar is also fine, the paper does not follow a storyline. This means that the paper is often too abrupt and the motivation of the research, the contributions and the rationale behind the study is also weak.

When considering the literature and references there is not enough content for the reader. Many concepts are left to the readers imagination and the research does not attract the reader.

Experimental design

The experimental design is acceptable but there is limited to no description of the dataset, its preprocessing. Thus the investigation is too limited and as a reviewer I cannot comment on its effectiveness for the wider audience. The research questions and the motives are not concrete so the reader is often lost.

According to the author, the dataset is real but does not have an appeal. The lack of details and reasoning effects the understandability of the paper.

Validity of the findings

Overall paper has a limited impact and needs significant extension to be accepted. In short when looking at the results the reader simply says "I am not surprised" thus the paper does not impress in terms of contributions and outcome.

Conclusion is acceptable but the abstract and introduction are too abrupt and do not follow a comfortable flow/ story.

Cite this review as

·

Basic reporting

• These suggestions are intended to refine the paper's language and align it with grammatical
conventions in the lines-110,114,123,134,153,160,172,175,214 so as to improve the overall readability
of the paper.
• Upon reviewing the paper, it is evident that the references majorly consist of older sources, with
relatively limited inclusion of more recent literature in a systematic manner.
• The structure of the article, figures, and tables are accepted.
• The results are not mentioned in a clear and precise manner.

Experimental design

•The author has performed rigorous investigation and methods are clearly defined.

Validity of the findings

• Additionally, there is room for improvement in the clarity and thoroughness with which the method
employed to achieve the reported accuracy rate is described.
• The evaluation methodology should be discussed intensely with a comparative analysis.
• The overall structure of the article can be improved and justified to enhance the clarity and precision of
the content.
• The conclusions are well-stated and related to the research.

Additional comments

• The overall structure of the article can be improved and justified to enhance the clarity and precision of
the content.

Reviewer 3 ·

Basic reporting

This was an interesting project, contributing to:
• Detection of suspicious activities and objects,
• Applied to data from a military installation,
• Using AI-tools and a
• Web-based display showing the status of items, cars, and people.
The 98.867% classification accuracy using a Decision Tree is impressive.

Basic Reporting

The article was generally professionally written easy to read and comprehend. The research applies A.I. enhanced detection software to produce a web-based, real-time display showing the status of suspicious items, cars, and people to improve national security, especially with the armed conflicts ongoing in Thailand’s southern border regions.

Experimental design

The research objective was to develop, test, and deliver a software detection system based on artificial intelligence for area surveillance, including buildings, roads, sites, objects, vehicles, and people. This detection program fuses and analyzes large amounts of data from various databases, allowing the integration of relationships between events to be visualized through web-based maps, graphs, and statistics. A unique aspect of this study is the focus on integrating news data, intending to soon implement rapid, real-time, big data analysis.

The research used real data for humans, vehicles, and items obtained from an Air Force Academy. The risk classifications are displayed on a web application. Access to this information is divided into two main user groups: regular users and administrators. Daily results provide statistics concerning objects, individuals, and vehicles through maps, statistical tables, and maps.

The research included assembly of the hardware and software for the detection tools and then using the data to train, validate, and test the detection model. Data pipelines were used to train automated models that extract, analyze, and store data from many sources.

Investigation Rigor and Methods

A critical part of the research was the evaluation of different technologies such as MySQL and Apache Hive, and Hadoop for processing these large datasets, comparing speed and capacity. Also, various machine learning algorithms, including ANN, SVM, k-NN, Naive Bayes, and Decision Tree, were compared for detection accuracy. Finally, the project produced a web application interface, discussed in the article, highlighting features and functions for accessing and analyzing the collected data.

A series of flow charts describe the overall system, and example output charts and maps are provided in the Figures. Accuracy comparisons are detailed in the included Tables. The data sets ranged from 10,000 to 5,000,000 records in CSV format.

For the overall project, a major purpose was to assess processing speed and the power of different technologies when dealing with massive amounts of data. The results showed that MySQL was the most stable for uses requiring immediate data retrieval from smaller data sets. As the data size exceeded 100,000 records, MySQL became unresponsive. The Apache Hive/Hadoop demonstrated better scalability and performed well with large data sets.

An additional set of experiments compared several machine-learning algorithms, including Artificial Neural Networks (ANN), Support Vector Machines (SVM), K-Nearest Neighbor (k-NN), Decision Tree, and Naive Bayes, implemented using Scikit-learn in Python. The results showed that Decision Tree had the best classification accuracy (98.867%) and the fastest prediction speed (0.005 milliseconds per sample).

Methods Description

The author details which databases, machine-learning algorithms, and web applications, and the use of data pipelines and implementations using Scikit-learn in Python. Several flow charts are included to explain how all the parts are assembled. However, I suspect anyone attempting to recreate this complex system would have to invest significant time and effort. Helpfully, the author provided many references.

Validity of the findings

The author applied a variety of performance, efficiency, computational usage and speed, and accuracy verification and validation experiments.

I didn’t find any reason to question the validity of the results. Underlying data was provided, but the data descriptions were in Thai. On the other hand, its probably not reasonable to expect big data files with millions of data point to be translated into English.

The manuscript lists descriptions of the types of data collected and analyzed. The charts provide some examples of statistical outputs from the data. But there is no detailed description of the data, the sources, nor whether any processing or testing of the raw data was performed other than the detection model’s output.

In addition, I’d have liked to see a discussion of how well Python performed in terms of stability, speed, and other aspects critical to implementation.

Additional comments

Although the narrative was informative and easy to read, some issues will need to be addressed.

I extracted the manuscript from the PDF, opened it in MS Word, and then reviewed it using MS Editor. The grammar, punctuation, and spelling editor flagged multiple errors, including:
1) Numerous extra letters with diacritic marks appeared throughout the article, especially before and after words or word strings, and often after commas. Unlike the Thai language, English and its characters do not use accents or diacritic marks, except for an occasional word from another Latin language. I have attached a marked-up copy of the original manuscript that flags many of the most obvious errors, but I probably missed some.

2) The use of the term Hadoop Hive is confusing. There are two different software programs available from Apache. Apache Hive is a data warehouse software project built on top of (integrated) Apache Hadoop. Hive is an SQL-like interface for querying databases and files. Apache Hadoop is a collection of utilities for networked computers, including the storage and processing of big data. I assume that is what the author was referring to: the combination of two Apache software tools.

3) The word suspicious is an English adjective. Its placement in phrases is often incorrect. Monitoring suspicious activity is a correct usage. Phrases such as suspicious detection tool, suspicious model, and suspicious tool all incorrectly imply that the detection tool or model is suspect or questionable. It’s the activity that is suspicious, not the detection tool, model, or program.

4) Many web interface charts have captions with Thai captions or a mix of English and Thai. Also, one of the charts shows location pins to illustrate detections using a color gradient to illustrate the degree of concern. I assumed the author used a 4-step high-to-low scale to detect suspicious activity. However, the descriptive labels for the colors include terms such as hard and mediam that confuse an English reader. Maybe the author meant medium. Hard could mean a firm detection, but it’s not at the top rank of the scale where I expected it to appear. Please clarify.

5) The descriptors in the raw data files are mainly in the Thai language and Thai script characters, so I could not address the data quality. Given the massive size of some data sets, it would be unreasonable to expect the author to translate it into English. Much of this data appears to be obtained from devices such as security cameras and news videos.

6) The narrative lacks descriptions of the specifics of the data other than a military college provided it. The types of data and the raw sources are illustrated in the Figures. I’d like to see a discussion in the text of the data sources, how the data was converted to a CSV format, any statistical analyses of the data for normalcy accuracy, and any checks to address the validity of the raw data.

7) I could not find any detailed descriptions of the statistics used for the performance comparisons or the statistical significance of the results. Maybe these are part of the software programs, but I’d expect the author to describe how the best-performing algorithm was determined other than it yielded the highest percent accuracy.

Annotated reviews are not available for download in order to protect the identity of reviewers who chose to remain anonymous.
Cite this review as

---

## Round 0.2 · Minor Revisions

There are a few minor comments that require your attention. Please address and resubmit the revised version along with the review response file.

Reviewer 1 ·

Basic reporting

The quality of the manuscript has improved since the first review. The flow is better and the readability has improved.

Experimental design

No concern

Validity of the findings

No concern

Additional comments

Accept.

Cite this review as

·

Basic reporting

Your idea is good and it is easily understandable.So make sure your content must be in professional English language.Latest paper was surveyed by the author for eg:NuxtJS Nuxt - The Intuitive Vue Framework.î https://nuxtjs.org (accessed Mar. 19,
328 2023).Quality of the content is predominant. Yes, Figures are relevant, high quality, well
labelled & described.Raw data supplied was limited Author have used upto 5mb, why? justtify your answer.

Experimental design

Experimental design was within the scope of the journal.
Author have identified the knowledge gap.
Author have to improve the technical and ethical standard.

Validity of the findings

Author have defined using machine learning algorithms the decision tree have reached with 98.867% accuracy.
Statistically not sound which has less number of data.
Conclusion are well stated.

Reviewer 3 ·

Basic reporting

The authors did a great job cleaning up the language, symbols, naming, etc. I ran my document editor and it found no significant errors.

I still find the title confusing "Enhancing Suspicious Activity Monitoring System with
AI-Based and Big Data Fusion"

I suggest that the authors consider: "Enhancing Monitoring of Suspicious Activities with AI-Based and Be Data Fusion"

I do not think that the authors mean to "Enhance Suspicious Activity"

Experimental design

The design remains as in the original, with the names of some of the software programs corrected.

Validity of the findings

I leave this question to my fellow reviewers. I see that they asked for some clarifications.

Additional comments

The authors clearly made a concerted effort to respond to my comments. If they did the same for the comments of the other reviewers, then I would recommend publication.

Cite this review as

---

## Round 0.3 · accepted · Accept

Congratulations, the revised version of the manuscript is satisfactory and it is recommended for publication.